# Semi-Supervised Deep Learning Model for Efficient Computation of Optical Properties of Suspended-Core Fibers

**DOI:** 10.3390/s22186751

**Published:** 2022-09-07

**Authors:** Gao Wang, Sufen Ren, Shuna Li, Shengchao Chen, Benguo Yu

**Affiliations:** 1State Key Laboratory of Dynamic Measurement Technology, North University of China, Taiyuan 030051, China; 2School of Information and Communication Engineering, Hainan University, Haikou 570228, China; 3School of Information and Communication Engineering, North University of China, Taiyuan 030051, China; 4School of Biomedical Information and Engineering, Hainan Medical University, Haikou 571199, China

**Keywords:** suspended-core fiber, inverse design, semi-supervised deep learning, optical properties

## Abstract

Suspended-core fibers (SCFs) are considered the best candidates for enhancing fiber nonlinearity in mid-infrared applications. Accurate modeling and optimization of its structure is a key part of the SCF structure design process. Due to the drawbacks of traditional numerical simulation methods, such as low speed and large errors, the deep learning-based inverse design of SCFs has become mainstream. However, the advantage of deep learning models over traditional optimization methods relies heavily on large-scale a priori datasets to train the models, a common bottleneck of data-driven methods. This paper presents a comprehensive deep learning model for the efficient inverse design of SCFs. A semi-supervised learning strategy is introduced to alleviate the burden of data acquisition. Taking SCF’s three key optical properties (effective mode area, nonlinear coefficient, and dispersion) as examples, we demonstrate that satisfactory computational results can be obtained based on small-scale training data. The proposed scheme can provide a new and effective platform for data-limited physical computing tasks.

## 1. Introduction

Suspended-core fiber (SCF) is a microstructured fiber (MOF) that is considered an excellent alternate choice for enhancing the nonlinear properties of optical fibers [1,2]. Compared to other microstructured fibers, SCF is easier to fabricate and can achieve higher numerical apertures (NA). Therefore, it is widely used in various fiber laser applications, such as Raman fiber lasers [3], Brillouin fiber lasers [4], etc. In addition, its application is also extended to high-capacity networks [5].

Accurate modeling and optimization of SCF structures usually rely on numerical computational methods such as finite difference, finite element methods [6], block iterative frequency domain methods [7]. Plane wave expansion [8]. However, large-scale iterative analysis is often required to obtain more accurate results in modeling and optimizing fiber optic structures. In addition, the large number of structural design parameters makes each round of iterative analysis time-consuming, and complex structures require multiple simulations to optimize the design. These requirements seriously reduce the efficiency of numerical methods and pose a serious challenge to traditional modeling and optimization methods.

With the rapid development of artificial intelligence (AI), especially the gradual improvement of deep learning (DL) techniques, researchers are actively seeking to apply DL techniques to solve challenging tasks, including materials science [9], chemistry [10], laser physics [11], particle physics [12], and quantum mechanics [13]. DL model that, as a typical data-driven approach, can learn the respective complex nonlinear relationships in a given dataset and abstract their relationships to form a solution strategy through a computational model consisting of multiple layers of data processing units. This avoids the high cost and inefficiency caused by human intervention in solution computation and the direct interaction between the fundamental physical laws in traditional optimization. As DL techniques spread in various fields, research on optics or optical systems based on DL has been explored and refined. On the one hand, a well-trained DL model can be utilized as a fast solver to predict a physical quantity, such as in fiber-optic demodulation systems [14,15], optical computational imaging [16], and biomedical engineering [17]. In addition, it can be seen as an optimization tool without expert empirical intervention for applications in the inverse design of optical materials, such as metamaterials [18,19,20], integrated photonics [21], plasmonics [22,23,24], etc.

DL techniques have now been widely applied to the problem of solving optical properties during the inverse design of MOFs. Da Silva Ferreira et al. combined a multilayer perceptron (MLP) and an extreme learning machine-artificial neural network (ELM-ANN) to calculate photonic crystal (PCs) for dispersion relations [25]. Chugh et al. introduced an ANN model to achieve fast computation of various optical properties of PC fibers (PCF) [26]. Yuan et al. applied a back-propagation neural network (BPNN) to accelerate the calculation process of SCF’s optical properties [27]. Without exception, however, the premise that these works guarantee superior performance requires a large-scale a priori data set to train the model. However, collecting and labeling the dataset is laborious and tedious. Therefore, exploring an effective deep learning model that can mine the nonlinear physical relationship between SCF geometric and optical properties in minimal data, is crucial in the reverse design process of MOFs.

In this paper, we proposed an efficient deep learning model for optimizing the optical parameter solving process for the inverse design of SCF structures with limited available a priori data. The proposed model consists of a Generative Adversarial Network (GAN) for data augmentation, and a back-propagation neural network (BPNN) cascaded for optical properties computation. The model employs a semi-supervised learning strategy that performs online data augmentation during training based on pre-collected data input to the computation model. The computation model calculates the optical properties of the designed SCF structure, including the effective mode area, nonlinear system, and dispersion. Experimental results show that the proposed model can obtain efficient and highly accurate optical property calculations with extremely limited a priori data. In conclusion, the model creates a new framework and platform for the efficient design of optical fiber structures with arbitrary microstructures and also provides reliable support for DL-based physical computational problems under minimal data sets.

This work is organized as follows. Section 2 describes the design and theoretical analysis of the SCF model in this work. Section 3 describes the structure and configuration of the proposed semi-supervised deep learning model for computing the optical properties of the SCF. Section 4 compares the predicted and actual values of the optical properties. Section 5 concludes this work.

## 2. Suspended-Core Fiber Design and Theoretical Analysis

### 2.1. Structure Design of SCFs

The structures of the SCFs used in this work are reproduced from reference [28]. Figure 1a–c show the cross-sectional schematics of the three-, four-, and six-bridges (cantilever) SCFs, respectively, and the corresponding Figure 1c shows the fundamental optical mode field distributions of the three-, four-, and six-bridges (cantilever) SCFs. The diameter, width of the cantilever, and the number of the cantilever of SCFs structure are denoted by *d*, *W*, and *n*, respectively.

### 2.2. Basic Theoretical Analysis

The real part of the RI of the chalcogenide materials can be calculated from Sellmeier equation as shown in Equation (Equation 1) [29].
(1)$$n^{2}\left(\lambda \right)=1+\sum_{i}\frac{A_{i}\lambda^{2}}{\lambda^{2}-\lambda_{i}^{2}},$$
where λ denotes the wavelength in free space, $A_{i}$ and $\lambda_{i}^{2}\left(i=1,2,3\right)$ are the constant of substrate materials of SCF. $As_{2}S_{3}$ was used as the core filling material for SCFs, and its material constants: $A_{1}$, $A_{2}$, $A_{3}$, $\lambda_{1}^{2}$, $\lambda_{2}^{2}$ and $\lambda_{3}^{2}$ are 1.8983, 1.9222, 0.8765, 0.0225, 0.0625 and 0.1225, respectively.

The effective index of the fundamental mode propagation of the SCF optical field distribution can be obtained by a finite element model (FEM). With the known effective index ($n_{eff}$), the dispersion of the SCF as a function of wavelength can be expressed by Equation (Equation 2).
(2)D(λ)=−λcd2dλ2Reneff(λ),
where Reneff(λ) is the real part of the effective index, and *c* is the light velocity in free space.

The nonlinear coefficient γ can be calculated by Equation (Equation 3) [29].
(3)$$\gamma =\left(2\pi /\lambda \right)\cdot \left(n_{2}/A_{\mathrm{eff}}\right),$$
where *c* is the light speed in free space, $n_{2}$ is the nonlinear refractive index and $n_{2}$ of $As_{2}S_{3}$ is $4.2\times 10^{-18}\;m^{2}/W$, the $A_{eff}$ denotes the effective area of the propagation mode in the SCF and is defined as Equation (Equation 4) [27].
(4)$$A_{\mathrm{eff}}={\left({\;\int \int \;}_{\infty }{\left|E\left(x,y\right)\right|}^{2}\;dx\mathrm{dy}\right)}^{2}/{\;\int \int \;}_{\mathrm{NLR}}\left|E\left(x,y\right)\right|{}^{4}\;dx\mathrm{dy},$$
where E(x,y) is the distribution of the optical field across the SCF cross-section, and NLR stands for nonlinear material region.

## 3. Semi-Supervised Deep Learning Model with Suspended-Core Fiber

### 3.1. Data Pre-Processing

The data pre-processing is necessary to avoid the problem of slow convergence before the training dataset is fed into the neural network model. The ’Min-Max’ normalization was used to linearly transform the original data and map the original data to the interval [0, 1], which can eliminate the adverse effects caused by singular samples in the characteristic data and improve the convergence speed and performance of the model. The transformation function can be defined as Equation (Equation 5).
(5)$$x^{*}=\frac{x-min(x)}{max(x)-min(x)}$$
where *x* represents the original eigenvalue of SCF and $x^{*}$ represents the normalized result.

### 3.2. Model Design for Optical Properties Calculation

The semi-supervised deep learning model used to compute the optical properties of the SCF is shown in Figure 1. The data processing process of the collation model is shown in Figure 2a, where the original data set is input to the GAN after the data mentioned above pre-processing, and the network generates a series of generative data with a distribution close to the original data. The structure of the GAN is shown in Figure 2b, which is an unsupervised deep learning model for generating data, consisting of a separate generative model (*G*) and discriminative model (*D*). The objective of the *G* is to generate actual data to deceive the *D* as much as possible, while the objective of the *D* is to distinguish the data generated by the *G* from the actual data as much as possible. The training process is a dynamic game process. The original data, instead of random noise, was used as the output of the *G* to make the generated data more realistic. The original data is fed to the *D* together with the generated data. The *G* is a typical encoder-decoder structure. In contrast, the *D* outputs a data truth score to encourage the *G* to generate actual data as much as possible, and their structures are shown in the red dashed box in Figure 2b. Then the generated data were fed to the BPNN for computation with the original data, and the network structure is shown in Figure 2c.

GAN and BPNN are entirely independent in the training phase, where the training process of GAN is a “Min-Max” dynamic game process. The loss function of the network can be established as Equation (Equation 6), while the training process of BPNN is an error back propagation process, and MSE is chosen as its loss function, which can be defined as Equation (Equation 7).
(6)$$Loss_{GAN}=\min_{G}\max_{D}E_{x∼p_{\mathrm{data}\;}}\left[\log D\left(x\right)\right]+E_{z∼p_{z}}\left[\log\left(1-D\left(G\left(z\right)\right)\right)\right],$$
(7)$$Loss_{BPNN}=\frac{1}{n}\sum_{i=1}^{n}{\left(y_{i}-\hat{y_{i}}\right)}^{2},$$
where *Z* is the random noise fed into the *G*, *n* represents the total number of samples, and $y_{i}$ is the predicted value of the neural network model, $\hat{y_{i}}$ is the actual value.

To ensure that the parameters are approximate to nonlinear functions in the error propagation process, *Sigmoid* [30] was used as the activation function between hidden layers, and the function can be formulated as Equation (Equation 8):(8)$$Sigmoid\left(x\right)=\frac{1}{1+e^{-x}},$$
where *x* is the input of the hidden node.

## 4. Experiments

The number of cantilevers of the SCFs used in the experiments was 3, 4, and 6, and each SCF contained 14 different core radii (0.4 μm, 0.45 μm, 0.5 μm, 0.55 μm, 0.6 μm, 0.7 μm, 0.75 μm, 0.8 μm, 0.85 μm, 0.9 μm, 0.95 μm, 1 μm, 1.05 μm, 1.1 μm) and 6 cantilever widths (0.6 μm, 0.7 μm, 0.8 μm, 0.9 μm, 1.0 μm, 1.1 μm). To build the dataset, the relevant optical features were calculated using a permutation form (e.g., *n* = 3, *r* = 0.4 μm, *W* = 0.07 μm). The dataset is divided into training, validation, and test sets in the ratio of 6:2:2 with no overlap between the three. During the training process, the learning rates were adjusted to 0.01, 0.005, and 0.001, respectively, and SGD, Adam, and Adagrad were selected as optimizers to explore the appropriate training configuration (e.g., Optimizer: Adam, Learning Rate: 0.001) to ensure the performance of the model. MSE was used as the loss function during model training, and MSE, RMSE, and MAE were chosen as error evaluation metrics to comprehensively assess the model’s effectiveness, which can be formulated as Equation (Equation 9), Equation (10) and Equation (Equation 11), respectively.
(9)$$MSE=\frac{1}{n}\sum_{i=1}^{n}{\left(y_{i}-\hat{y_{i}}\right)}^{2},$$
(10)$$RMSE=\sqrt{\frac{1}{n}\sum_{i=1}^{n}{\left(y_{i}-\hat{y_{i}}\right)}^{2},}$$
(11)$$MAE=\frac{1}{n}\sum_{i=1}^{n}\left|\hat{y_{i}}-y_{i}\right|,$$
where *n* represents the total number of samples, $y_{i}$ is the predicted value, and $\hat{y_{i}}$ denotes the actual value.

### 4.1. Model Training Process

In the training process, MSE was used as the loss function. We calculated the MSE of the training and verification process of different learning rates and optimizers (SGD, Adagrad, Adam), as shown in Table 1. The similar MSEs on the validation dataset indicate that the models trained by all three optimizers can achieve satisfactory results. Detailed prediction results and analysis of the effective mode area, nonlinear coefficients, and dispersion of the SCF will be presented in the following.

### 4.2. Effective Mode Area

The waveguide characteristics played by SCF in mid-infrared applications depend mainly on its effective mode area. This relationship can be seen in Equation (Equation 3). The smaller its mode area, the greater the advantage of the fiber in nonlinear applications. Conversely, a larger effective mode area can also enhance the effectiveness of SCF in optical power transmission applications.

This section shows the proposed model’s results in predicting the SCF’s effective mode area ($A_{eff}$). The predicted and actual values for each sample in the test dataset are compared, and the performance of the obtained model with different training configurations is analyzed, as shown in Table 2. Among them, the SGD optimizer and the Adam optimizer have better performance.

Based on the above conclusions, the models trained by Adam and SGD were selected for testing to compare the error between the actual and predicted values of $A_{eff}$, as shown in Figure 3. As these points get closer to *y* equal to 0, the residual between the predicted and actual values is more diminutive. Their proximity to the origin is positively correlated with the model’s performance. The model with Adam as the optimizer works best in the effective pattern area prediction for SCF when the learning rate is 0.001. This means that the best-performing model can be obtained with this set of training configurations (Optimizer: Adam, LR: 0.001).

Figure 4, Figure 5 and Figure 6 show the predicted values of $A_{eff}$ during wavelength variation compared to the actual values for models trained by different optimizers for a different number of cantilevers, core radii, and cantilever widths. The $A_{eff}$ corresponding to these parameters has never been recorded or provided. Figure 4, Figure 5 and Figure 6 show the $A_{eff}$ predictions of the model when the number of cantilevers is 3, 4, and 6, respectively. As can be seen in the $A_{eff}$ prediction results for the SCF with several cantilevers of 3, the network trained by the Adam optimizer predicts best when the wavelength is between 3 μm and 4 μm. When the number of cantilever beams is 4, the predicted value of $A_{eff}$ by the network trained by Adam is closer to the actual value in the wavelength range of 1 μm–2 μm. When the number of cantilever beams is 6, in the wavelength range of 2 μm to 3 μm, the best network prediction trained by Adam is still obtained.

Combined with Figure 4, Figure 5 and Figure 6, the predicted value of $A_{eff}$ in the test dataset of Adam trained model is closer to the real value. This can be proved by the quantitative analysis in Table 3 that our proposed model can work well in predicting $A_{eff}$ of SCF.

### 4.3. Nonlinear Coefficient

This section uses the well-trained model to predict the nonlinear coefficient (γ) directly related to the performance presented by the SCFs in nonlinear applications. Table 4 shows the quantitative analysis of the prediction results for the γ value. The model trained with the SGD optimizer and Adagrad optimizer could achieve better results. Figure 7 shows the magnitude of the residuals between the predicted and actual γ values of the test samples for the models trained with different LRs by the SGD optimizer and the Adgrad optimizer. The γ predicted values of the SGD-trained model are closest to the actual values. Combined with Table 4 and Figure 7, the model trained by the SGD optimizer with 0.001 LR shows the best results in terms of evaluation metrics, which means that the trained model configured by this parameter achieves superior performance in terms of γ prediction.

Figure 8, Figure 9 and Figure 10 compare the predicted and actual values of the γ value for different combinations of the number of cantilevers, core radius, and cantilever width for the models trained by different optimizers during the wavelength variation. Figure 8, Figure 9 and Figure 10 show the comparison results when the number of cantilever beams is 3, 4, and 6, respectively. The γ values gradually decrease as the wavelength increases, and the predicted values of γ of the network model trained by SGD are closer to the actual values. In addition, the network trained by SGD shows higher prediction performance than other networks in the wavelength range of 2 μm to 3 μm.

The proposed model can predict γ well during the linear variation of wavelength. Combining Figure 8, Figure 9 and Figure 10, the predicted values of γ in the SGD training model test dataset (compared with other optimizers) are closer to the actual values. Moreover, the quantitative analysis in Table 5 proves that our proposed model can better predict the γ value of SCF.

### 4.4. Dispersion

Dispersion (*D*) is essential in generating SCF-based supercontinuum spectra. Table 6 shows the quantitative analysis of the models in terms of *D*-value prediction, where the models trained by SGD and Adam show good performance. Figure 11 shows the magnitude of the residuals between the predicted and actual values of *D* in the test set of the model trained by the SGD optimizer and Adam optimizer at different LRs. The model trained by Adam with an LR of 0.001 achieves the best performance.

Figure 12, Figure 13 and Figure 14 compare the predicted and actual values of the *D* value for different combinations of the number of cantilevers, core radius, and cantilever width for the models trained by different optimizers during the wavelength variation. Figure 12, Figure 13 and Figure 14 show the comparison results when the number of cantilevers is 3, 4, and 6, respectively. The network trained by the Adam optimizer makes a better prediction of the *D* value of the SCF as the wavelength varies linearly, implying that its prediction is closer to the actual value than the other two optimizers. However, during the model’s prediction of the *D* value, its predicted value is smoother with the wavelength, probably because the *D* value shows severe irregularities in the range of wavelength variation. The model is less sensitive to such nonlinear irregularities. Even though the proposed model can present superior prediction results, the problem still poses an obstacle to its excellent prediction results. To address this problem, it is straightforward to consider increasing the number of iterations and adjusting the nonlinear activation function to solve it.

Combining Figure 12, Figure 13 and Figure 14, we can see that the predicted values of *D* in the test dataset for the Adam-trained model (compared to the other optimizers) are much closer to the actual values. Combined with the quantitative analysis in Table 7, we can prove that the proposed model can predict *D* effectively.

### 4.5. Discussion

During the experiments, the LRs were adjusted to 0.01, 0.005, and 0.001. SGD, Adam, and Adagrad were selected as the optimizers to explore the model’s performance. Combined with the above experimental results, the Adam-based model has a good prediction effect on the $A_{eff}$ and *D*. The model with SGD optimizer is the best in the prediction process of γ. The LR impacted the model’s performance when well-trained models were used to predict optical properties. Table 8 shows the combined prediction of the model for the three optical properties (evaluated in terms of MSE) under different training configurations (optimizer and LR). The overall prediction performance of the model is best when the LR is 0.001. The proposed model can predict the important optical properties of SCF well. Furthermore, It is straightforward to consider increasing the feature dimensionality of the input, increasing the number of iterations, and using more significant augmented data to improve the model’s predictive performance further. However, these strategies will also inevitably increase resource utilization and iteration time.

## 5. Conclusions

In summary, this paper presents a comprehensive semi-supervised deep learning model that can efficiently solve important optical properties (effective mode area, nonlinear coefficients, and dispersion) in the inverse design of an SCF with an extremely limited a priori dataset. The model consists of a GAN, which is used to augment the limited a priori data set efficiently, and a BPNN cascade, which receives both the original and augmented data sets for calculating and optimizing the optical properties of the SCF. It is experimentally demonstrated that the model accurately predicts the optical properties. In addition, different training configurations (optimizer and learning rate) were used to train the model during the experiments to demonstrate the reliability of the model. The strategy can provide practical support for the inverse design of microstructured optical fibers that can be used for other materials and a new and reliable platform for machine learning-based physical property calculations with extremely limited a priori data sets.

## Figures and Tables

**Figure 1 sensors-22-06751-f001:**
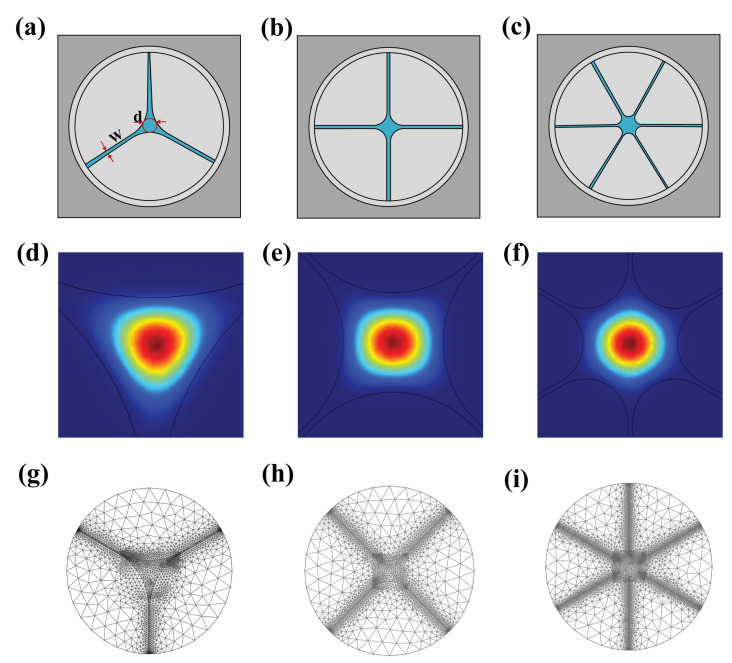
(**a**–**c**) Cross section of the core area for simulation with three bridges, four bridges, and six bridges, respectively. (**d**–**f**) Fundamental optical mode field distribution of three bridges, four bridges, and six bridges, respectively. (**g**–**i**) Triangular mesh of finite element for the three brigades, four bridges, and six bridges SCF, respectively.

**Figure 2 sensors-22-06751-f002:**
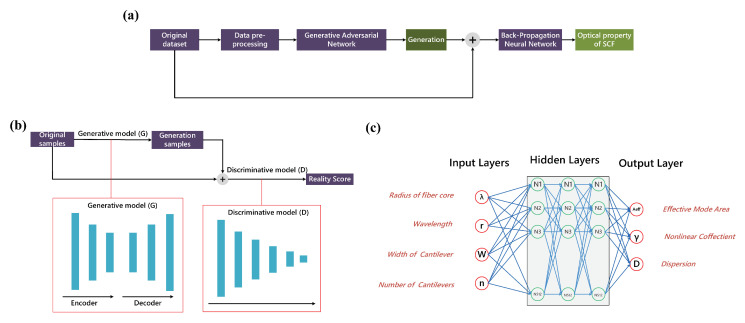
(**a**) Computational flow of the proposed model. (**b**) Processing flow and structure diagram of a generative adversarial network for data augmentation, G is a typical encoder-decoder structure with 512, 256, 128, 128, 256, 512 neurons per layer (full-connection layer, from left to right), and D with 512, 256, 128, 64, 32, 1 neuron per layer (full-connection layer, from left to the right). (**c**) The structure of the back-propagation neural network for optical properties calculation has three hidden layers with 512 neurons each, one input layer and one output layer, the input layer corresponds to the four geometric features of the SCF, and the output layer corresponds to the optical properties to be calculated.

**Figure 3 sensors-22-06751-f003:**
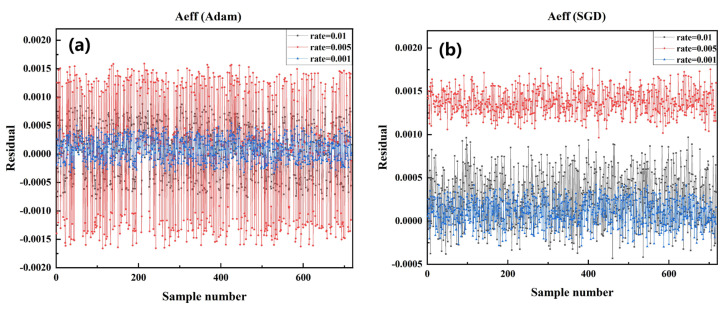
Comparison of the residual between the predicted value and the actual value of $A_{eff}$ in the test data set of the model trained by each algorithm under different learning rates in SGD optimizer and Adam optimizer. (**a**) Adam optimizer, LR: 0.01, 0.001, 0.005 (**b**) SGD optimizer, LR: 0.01, 0.001, 0.005.

**Figure 4 sensors-22-06751-f004:**
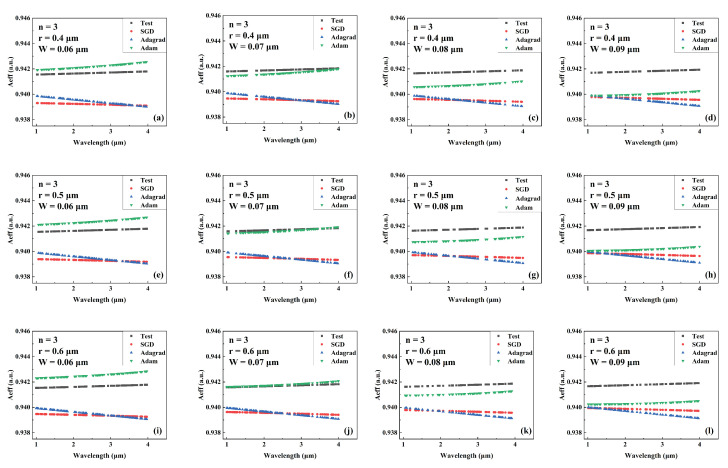
Comparison of the predicted and actual values of $A_{eff}$ in the test data set of the network model trained by different optimizers for different combinations of core radius and cantilever width (number of cantilevers: 3). (**a**) r = 0.4 μm, W = 0.06 μm. (**b**) r = 0.4 μm, W = 0.07 μm. (**c**) r = 0.4 μm, W = 0.08 μm. (**d**) r = 0.4 μm, W = 0.09 μm. (**e**) r = 0.5 μm, W = 0.06 μm. (**f**) r = 0.5 μm, W = 0.07 μm. (**g**) r = 0.5 μm, W = 0.08 μm. (**h**) r = 0.5 μm, W = 0.09 μm. (**i**) r = 0.6 μm, W = 0.06 μm. (**j**) r = 0.6 μm, W = 0.07 μm. (**k**) r = 0.6 μm, W = 0.08 μm. (**l**) r = 0.6 μm, W = 0.09 μm.

**Figure 5 sensors-22-06751-f005:**
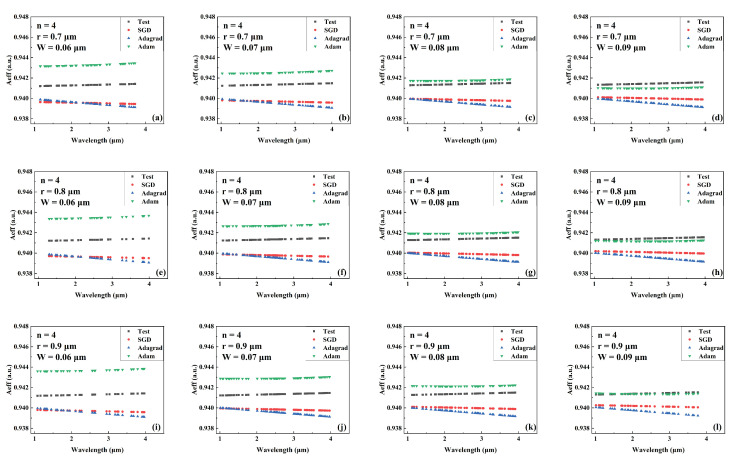
Comparison of the predicted and actual values of $A_{eff}$ in the test data set of the network model trained by different optimizers for different combinations of core radius and cantilever width (number of cantilevers: 4). (**a**) r = 0.7 μm, W = 0.06 μm. (**b**) r = 0.7 μm, W = 0.07 μm. (**c**) r = 0.7 μm, W = 0.08 μm. (**d**) r = 0.7 μm, W = 0.09 μm. (**e**) r = 0.8 μm, W = 0.06 μm. (**f**) r = 0.8 μm, W = 0.07 μm. (**g**) r = 0.8 μm, W = 0.08 μm. (**h**) r = 0.8 μm, W = 0.09 μm. (**i**) r = 0.9 μm, W = 0.06 μm. (**j**) r = 0.9 μm, W = 0.07 μm. (**k**) r = 0.9 μm, W = 0.08 μm. (**l**) r = 0.9 μm, W = 0.09 μm.

**Figure 6 sensors-22-06751-f006:**
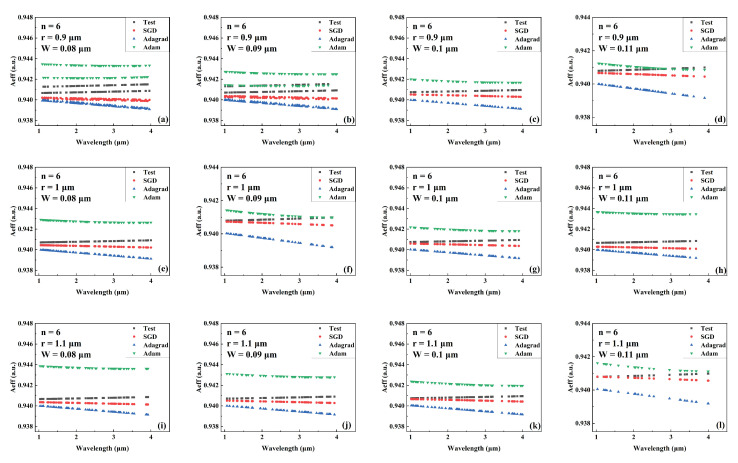
Comparison of the predicted and actual values of $A_{eff}$ in the test data set of the network model trained by different optimizers for different combinations of core radius and cantilever width (number of cantilevers: 6). (**a**) r = 0.9 μm, W = 0.08 μm. (**b**) r = 0.9 μm, W = 0.09 μm. (**c**) r = 0.9 μm, W = 0.1 μm. (**d**) r = 0.9 μm, W = 0.11 μm. (**e**) r = 1 μm, W = 0.08 μm. (**f**) r = 1 μm, W = 0.09 μm. (**g**) r = 1 μm, W = 0.1 μm. (**h**) r = 1 μm, W = 0.11 μm. (**i**) r = 1.1 μm, W = 0.08 μm. (**j**) r = 1.1 μm, W = 0.09 μm. (**k**) r = 1.1 μm, W = 0.1 μm. (**l**) r = 1.1 μm, W = 0.11 μm.

**Figure 7 sensors-22-06751-f007:**
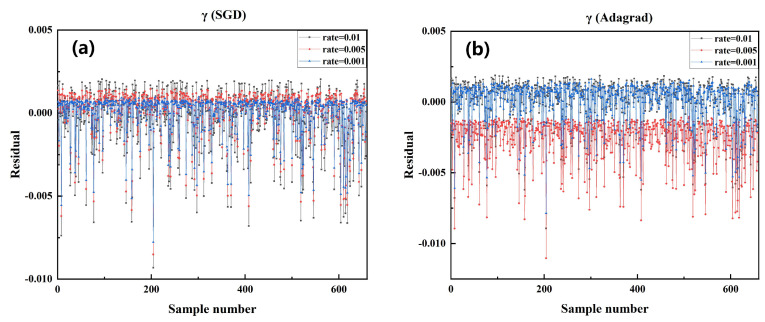
Comparison of the residual between the predicted value and the actual value of γ in the test data set of the model trained by each algorithm under different learning rates in SGD optimizer and Adam optimizer. (**a**) SGD optimizer, LR: 0.01, 0.001, 0.005 (**b**) Adagrad optimizer, LR: 0.01, 0.001, 0.005.

**Figure 8 sensors-22-06751-f008:**
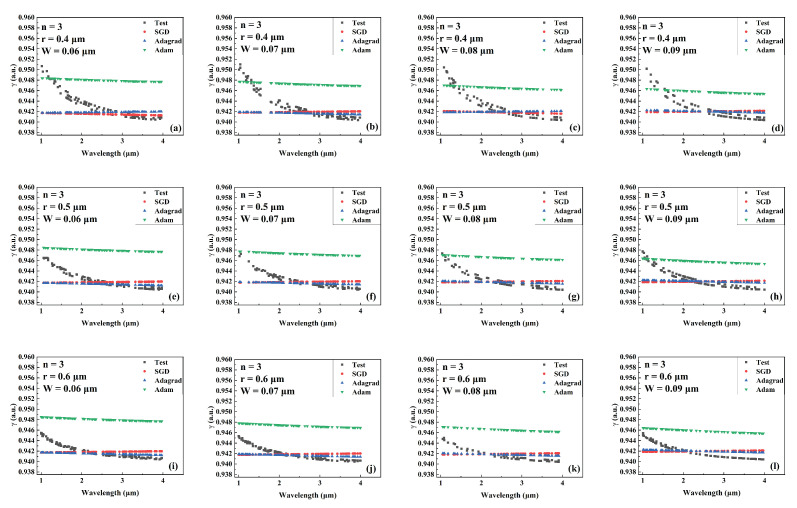
Comparison the predicted and actual values of γ in the test data set of the network model trained by different optimizers for different combinations of core radius and cantilever width (number of cantilevers: 3). (**a**) r = 0.4 μm, W = 0.06 μm. (**b**) r = 0.4 μm, W = 0.07 μm. (**c**) r = 0.4 μm, W = 0.08 μm. (**d**) r = 0.4 μm, W = 0.09 μm. (**e**) r = 0.5 μm, W = 0.06 μm. (**f**) r = 0.5 μm, W = 0.07 μm. (**g**) r = 0.5 μm, W = 0.08 μm. (**h**) r = 0.5 μm, W = 0.09 μm. (**i**) r = 0.6 μm, W = 0.06 μm. (**j**) r = 0.6 μm, W = 0.07 μm. (**k**) r = 0.6 μm, W = 0.08 μm. (**l**) r = 0.6 μm, W = 0.09 μm.

**Figure 9 sensors-22-06751-f009:**
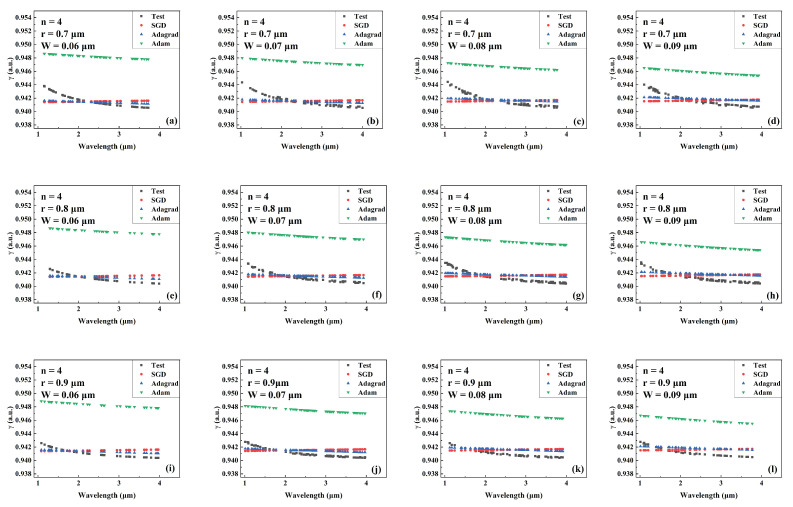
Comparision the predicted and actual values of γ in the test data set of the network model trained by different optimizers for different combinations of core radius and cantilever width (number of cantilevers: 4). (**a**) r = 0.7 μm, W = 0.06 μm. (**b**) r = 0.7 μm, W = 0.07 μm. (**c**) r = 0.7 μm, W = 0.08 μm. (**d**) r = 0.7 μm, W = 0.09 μm. (**e**) r = 0.8 μm, W = 0.06 μm. (**f**) r = 0.8 μm, W = 0.07 μm. (**g**) r = 0.8 μm, W = 0.08 μm. (**h**) r = 0.8 μm, W = 0.09 μm. (**i**) r = 0.9 μm, W = 0.06 μm. (**j**) r = 0.9 μm, W = 0.07 μm. (**k**) r = 0.9 μm, W = 0.08 μm. (**l**) r = 0.9 μm, W = 0.09 μm.

**Figure 10 sensors-22-06751-f010:**
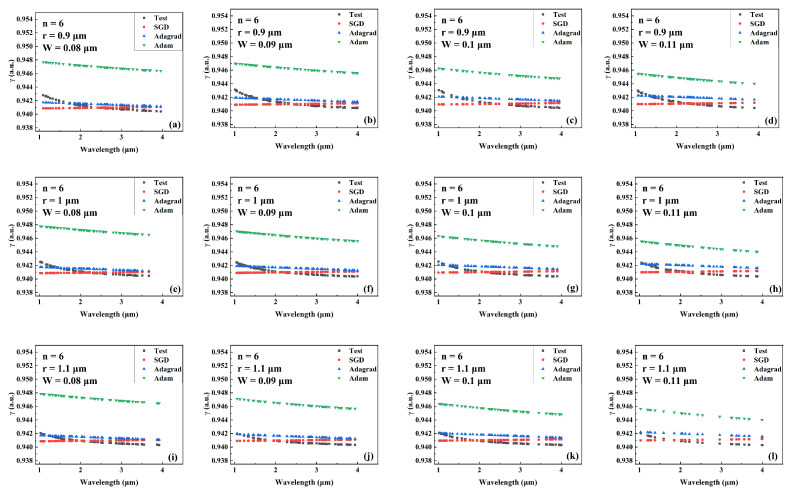
Comparison the predicted and actual values of γ in the test data set of the network model trained by different optimizers for different combinations of core radius and cantilever width (number of cantilevers: 6). (**a**) r = 0.9 μm, W = 0.08 μm. (**b**) r= 0.9 μm, W = 0.09 μm. (**c**) r = 0.9 μm, W = 0.1 μm. (**d**) r = 0.9 μm, W = 0.11 μm. (**e**) r = 1 μm, W = 0.08 μm. (**f**) r = 1 μm, W = 0.09 μm. (**g**) r = 1 μm, W = 0.1 μm. (**h**) r = 1 μm, W = 0.11 μm. (**i**) r = 1.1 μm, W = 0.08 μm. (**j**) r = 1.1 μm, W = 0.09 μm. (**k**) r = 1.1 μm, W = 0.1 μm. (**l**) r = 1.1 μm, W = 0.11 μm.

**Figure 11 sensors-22-06751-f011:**
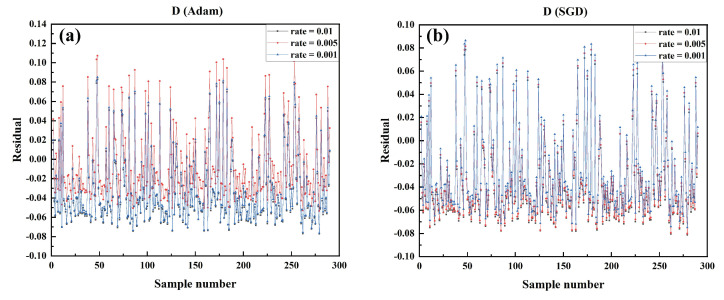
Comparison of the residual between the predicted value and the actual value of *D* in the test data set of the model trained by each algorithm under different learning rates in SGD optimizer and Adam optimizer. (**a**) SGD optimizer, LR: 0.01, 0.001, 0.005 (**b**) Adam optimizer, LR: 0.01, 0.001, 0.005.

**Figure 12 sensors-22-06751-f012:**
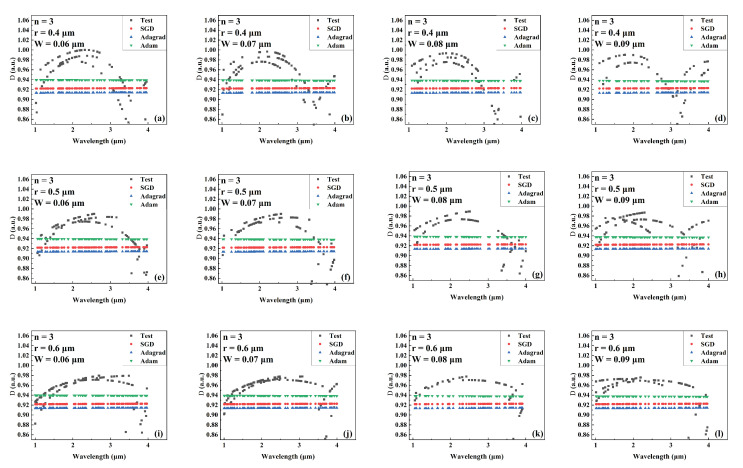
Comparison the predicted and actual values of *D* in the test data set of the network model trained by different optimizers for different combinations of core radius and cantilever width (number of cantilevers: 3). (**a**) r = 0.4 μm, W = 0.06 μm. (**b**) r = 0.4 μm, W = 0.07 μm. (**c**) r = 0.4 μm, W = 0.08 μm. (**d**) r = 0.4 μm, W = 0.09 μm. (**e**) r = 0.5 μm, W = 0.06 μm. (**f**) r = 0.5 μm, W = 0.07 μm. (**g**) r = 0.5 μm, W = 0.08 μm. (**h**) r = 0.5 μm, W = 0.09 μm. (**i**) r = 0.6 μm, W = 0.06 μm. (**j**) r = 0.6 μm, W = 0.07 μm. (**k**) r = 0.6 μm, W = 0.08 μm. (**l**) r = 0.6 μm, W = 0.09 μm.

**Figure 13 sensors-22-06751-f013:**
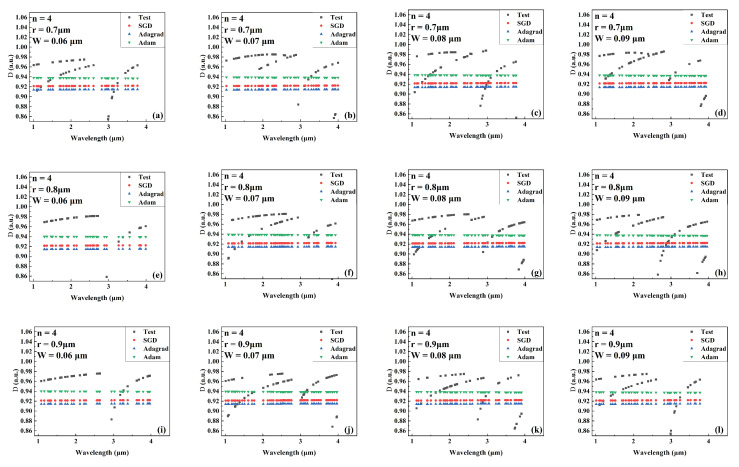
Comparison the predicted and actual values of *D* in the test data set of the network model trained by different optimizers for different combinations of core radius and cantilever width (number of cantilevers: 4). (**a**) r = 0.7 μm, W = 0.06 μm. (**b**) r = 0.7 μm, W = 0.07 μm. (**c**) r = 0.7 μm, W = 0.08 μm. (**d**) r = 0.7 μm, W = 0.09 μm. (**e**) r = 0.8 μm, W = 0.06 μm. (**f**) r = 0.8 μm, W = 0.07 μm. (**g**) r = 0.8 μm, W = 0.08 μm. (**h**) r = 0.8 μm, W = 0.09 μm. (**i**) r = 0.9 μm, W = 0.06 μm. (**j**) r = 0.9 μm, W = 0.07 μm. (**k**) r = 0.9 μm, W = 0.08 μm. (**l**) r = 0.9 μm, W = 0.09 μm.

**Figure 14 sensors-22-06751-f014:**
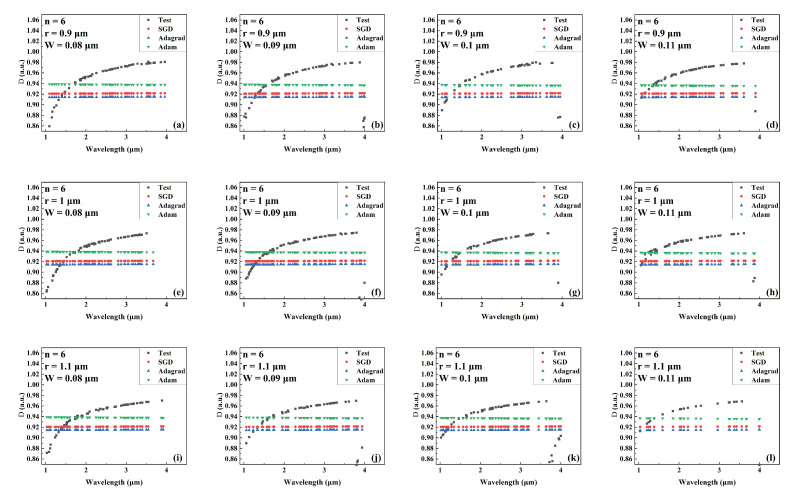
Comparision the predicted and actual values of *D* in the test data set of the network model trained by different optimizers for different combinations of core radius and cantilever width (number of cantilevers: 6). (**a**) r = 0.9 μm, W = 0.08 μm. (**b**) r = 0.9 μm, W = 0.09 μm. (**c**) r = 0.9 μm, W = 0.1 μm. (**d**) r = 0.9 μm, W = 0.11 μm. (**e**) r = 1 μm, W = 0.08 μm. (**f**) r = 1 μm, W = 0.09 μm. (**g**) r = 1 μm, W = 0.1 μm. (**h**) r = 1 μm, W = 0.11 μm. (**i**) r = 1.1 μm, W = 0.08 μm. (**j**) r = 1.1 μm, W = 0.09 μm. (**k**) r = 1.1 μm, W = 0.1 μm. (**l**) r = 1.1 μm, W = 0.11 μm.

**Table 1 sensors-22-06751-t001:** Average mean square deviation of a neural network model of training process and verification process under different learning rates and optimizers.

Optimizer	LR	MSE (Training, a.u.)	MSE (Validation, a.u.)
SGD	0.01	0.0061	0.0043
	0.005	0.0058	0.0043
	0.001	0.0064	0.0043
Adagrad	0.01	0.0054	0.0043
	0.005	0.0069	0.0046
	0.001	0.0008	0.0044
Adam	0.01	0.0108	0.0044
	0.005	0.0057	0.0046
	0.001	0.0052	0.0043

**Table 2 sensors-22-06751-t002:** Performance evaluation results for predictions of $A_{eff}$ on the test dataset with models trained with different training configuration.

Optimizer	LR	MSE (×$10^{-6}$, a.u.)	RMSE (a.u.)	MAE (a.u.)
SGD	0.01	1.63	0.0004	0.0003
	0.005	1.91	0.0014	0.0014
	0.001	3.78	0.0002	0.0002
Adagrad	0.01	2.38	0.0049	0.0049
	0.005	4.92	0.0022	0.0022
	0.001	8.01	0.0028	0.0028
Adam	0.01	2.17	0.0005	0.0004
	0.005	1.13	0.0011	0.0009
	0.001	0.51	0.0002	0.0002

**Table 3 sensors-22-06751-t003:** Average of each model on evaluation metrics when predicting $A_{eff}$ in the test dataset under different combinations of parameters.

Optimizer	MSE (×$10^{-6}$, a.u.)	RMSE (a.u.)	MAE (a.u.)
SGD	0.0704	0.00066	0.00062
Adagrad	5.10	0.0033	0.0033
Adam	0.467	0.00058	0.0005

**Table 4 sensors-22-06751-t004:** Performance evaluation results for predictions of γ on test datasets with models trained with different optimizers and different LRs.

Optimizer	LR	MSE (×$10^{-6}$, a.u.)	RMSE (a.u.)	MAE (a.u.)
SGD	0.01	2.87	0.0017	0.0012
	0.005	1.85	0.0014	0.0010
	0.001	1.2	0.0011	0.0007
Adagrad	0.01	2.39	0.0015	0.0011
	0.005	7.82	0.0028	0.0024
	0.001	1.66	0.0013	0.0010
Adam	0.01	6.92	0.0026	0.0025
	0.005	8.06	0.0028	0.0025
	0.001	4.37	0.0021	0.0016

**Table 5 sensors-22-06751-t005:** Average of each model on evaluation metrics when predicting γ in the test dataset under different combinations of parameters.

Optimizer	MSE (×$10^{-6}$, a.u.)	RMSE (a.u.)	MAE (a.u.)
SGD	1.97	0.0014	0.001
Adagrad	3.96	0.0019	0.0015
Adam	6.45	0.0025	0.0022

**Table 6 sensors-22-06751-t006:** Performance evaluation results for predictions of *D* on the test dataset with models trained with different optimizers and different LRs.

Optimizer	LR	MSE (a.u.)	RMSE (a.u.)	MAE (a.u.)
SGD	0.01	0.01366	0.11686	0.06026
	0.005	0.01421	0.11920	0.08287
	0.001	0.01349	0.11614	0.06438
Adagrad	0.01	0.01455	0.12061	0.05501
	0.005	0.01350	0.11617	0.06380
	0.001	0.01367	0.11692	0.07498
Adam	0.01	0.01354	0.011638	0.06264
	0.005	0.01347	0.11604	0.06417
	0.001	0.01345	0.11602	0.06414

**Table 7 sensors-22-06751-t007:** Average of each model on evaluation metrics when predicting *D* in the test dataset under different combinations of parameters.

Optimizer	MSE (a.u.)	RMSE (a.u.)	MAE (a.u.)
SGD	0.01378	0.11740	0.06917
Adagrad	0.01390	0.11790	0.06460
Adam	0.01350	0.11617	0.06445

**Table 8 sensors-22-06751-t008:** MSE averages for models trained using different training configurations (optimizer and LR) predict different optical properties, the bold denotes the optimal in this prediction.

Optimizer	LR	$A_{\mathrm{eff}}$ (× $10^{-6}$)	γ (× $10^{-6}$)	*D* (× $10^{-2}$)
SGD	0.01	1.63	2.87	1.366
	0.005	1.91	1.85	1.421
	0.001	3.78	**1.2**	1.349
Adagrad	0.01	2.38	2.39	1.455
	0.005	4.92	7.82	1.350
	0.001	8.01	1.66	1.367
Adam	0.01	2.17	6.92	1.354
	0.005	1.13	8.06	1.347
	0.001	**0.51**	4.37	**1.345**

## Data Availability

Not applicable.

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
