# Peer review of "Semi-Supervised Deep Learning Model for Efficient Computation of Optical Properties of Suspended-Core Fibers"

_sensors, 2022, doi:10.3390/s22186751_

Round 1

Reviewer 1 Report

The manuscript proposes a novel semi-supervised deep-learning method to predict some key properties of suspended core fibers for nonlinear applications, in a well-structured presentation. The work solves the problem of big training datasets needed in similar works (e.g. [27] in the manuscript). I would support the publication of the manuscript in Sensors, but still with some questions to be answered:

1. The outputs include effective mode area and nonlinear coefficient. Now that the deep learning model can predict the former, the nonlinear coefficient could be explicitly calculated using Eq. (3) of the manuscript. And vice versa. Then why are these two, which seem dependent, studied in separate sessions?

2. Will the length of cantilevers make a difference to the training result? Of course, when it is large, its influence may be minimal. Is it possible to study the problem for smaller values?

3. The figure quality demands improvement. In particular, the figures showing the comparison of precited and actual values (e.g. Figs. 4-6) are unreadable in resolution. In Fig. 1 (d-f), the mode profiles need to be produced with a higher mesh when simulated. 

4. The tables showing performance evaluation results need to indicate the unit used to calculate MSE, RMSE and MAE. Thus, readers can understand better the performances.

Author Response

We gratefully thank the editor and all reviewers for their time spent making their constructive remarks and valuable suggestions, which have significantly raised the quality of the manuscript and enabled us to improve it. The reviewers' suggested revisions and comments were accurately incorporated and considered. Below the reviewers' comments are response point by point, and the revisions are indicated. Responses to reviewer 1's comments are in the attachment.

Reviewer 2 Report

In this manuscript, the authors presented a semi-supervised deep learning model which can efficiently solve important optical properties (effective mode area, nonlinear coefficients, and dispersion) in the inverse design of an suspended-core fibers with an extremely limited a priori dataset.  The presented work is very useful for optical fiber design.

A few comments are listed as follows:

1. The tense should be consistent in the paper. Either use present tense or past tense throughout the paper when describing the work. Mixing tenses should be avoided.

2. Resolution of most of the figures is too low.

3. I suggest adding some discussion or explanation for section 4.1.

4. In section 4.2, “Among them, the degenerate optimizer and the Adam optimizer have better performance” . In Table 2,  SGD and Adam have better performance. To avoid confusion, I suggest keeping consistent for the optimize names.

5. Figure 3 shows the comparison  results between Adam and SGD, however, in the third paragraph, “the models trained by Adam and Adagrad were selected for testing to compare…”. Which one is correct, please double confirm.

6.  The three parameters for fiber design, i.e., effective mode area, nonlinear coefficient, and dispersion, the Optimizers are Adam, SGD and Adam, respectively. I suggest adding a table or conclusion to make the result more clear.

I recommend this paper for major revision.

Author Response

We gratefully thank the editor and all reviewers for their time spent making their constructive remarks and valuable suggestions, which have significantly raised the quality of the manuscript and enabled us to improve it. The reviewers' suggested revisions and comments were accurately incorporated and considered. Below the reviewers' comments are response point by point, and the revisions are indicated. Responses to reviewer 2's comments are in the attachment.

Round 2

Reviewer 2 Report

The authors have addressed to the raised concerns and have implemented all the suggested changes.